# Understanding the adsorption process in ZIF-8 using high pressure crystallography and computational modelling

Claire L. Hobday [1], Christopher H. Woodall[2], Matthew J. Lennox [3], Mungo Frost[2], Konstantin Kamenev [2], Tina Düren [3], Carole A. Morrison [1] & Stephen A. Moggach[1]

Some porous crystalline solids change their structure upon guest inclusion. Unlocking the potential of these solids for a wide variety of applications requires full characterisation of the response to adsorption and the underlying framework–guest interactions. Here, we introduce an approach to understanding gas uptake in porous metal-organic frameworks (MOFs) by loading liquefied gases at GPa pressures inside the Zn-based framework ZIF-8. An integrated experimental and computational study using high-pressure crystallography, grand canonical Monte Carlo (GCMC) and periodic DFT simulations has revealed six symmetry-independent adsorption sites within the framework and a transition to a high-pressure phase. The cryogenic high-pressure loading method offers a different approach to obtaining atomistic detail on guest molecules. The GCMC simulations provide information on interaction energies of the adsorption sites allowing to classify the sites by energy. DFT calculations reveal the energy barrier of the transition to the high-pressure phase. This combination of techniques provides a holistic approach to understanding both structural and energetic changes upon adsorption in MOFs.

[1] EaStChem School of Chemistry and Centre for Science at Extreme Conditions, University of Edinburgh, David Brewster Road, Joseph Black Building, Edinburgh EH9 3FJ, UK. [2] Department of Engineering, Centre for Science at Extreme Conditions, University of Edinburgh, Edinburgh EH9 3FD, UK. [3] Centre for Advanced Separations Engineering, Department of Chemical Engineering, University of Bath, Bath BA2 7AY, UK. Correspondence and requests for materials should be addressed to C.L.H. (email: clh65@bath.ac.uk) or to T.Dür. (email: t.duren@bath.ac.uk) or to S.A.M. (email: s.moggach@ed.ac.uk)

Understanding how porous metal organic frameworks (MOFs) interact with guest molecules of commercial interest, and how the framework structure adapts to the presence of these guest molecules, is vital for the further development and commercialisation of MOFs in a wide range of applications[1–3]. Despite a vast library of synthesised and characterised MOFs, only a small percentage of these frameworks have been investigated extensively to correlate their structure and adsorption properties[4]. In the MOF subset of the CSD, which contains all of these structures, there are 6, 18, 24, and 23 single crystal structures of MOFs containing modelled Ar, $CH_4$, $N_2$ and $O_2$, respectively—a total of 77 structures. Of these, only 17 were determined at room temperature –3, 9, 3, and 2 for Ar, $CH_4$, $N_2$, and $O_2$, respectively (see Supplementary Fig. 1)[4]. Whilst reports using techniques such as IR[5,6], Raman[7] and solid-state NMR spectroscopy[8] have inferred structural changes on the binding modes of gases within MOFs, more direct evidence of structural changes can be obtained using in-situ crystallography. Such experiments usually involve a gas capillary cell, where MOFs can be exposed to varying pressures of gas. This equipment is typically housed at dedicated central facilities in order to achieve the highest resolution for the position of guest molecules in the pores[9–11]. The prototypical MOF chosen for the first ever gas capillary cell experiment was ZIF-8 ($Zn_6(mIm)_{12}$, mIm = 2-methylimidazole)[12,13]. The structure contains one central nanopore per unit cell, with a volume of ≈2500 $Å^3$ and pore diameter of 11.6 Å[14]. Connecting these large nanopores are eight six-membered ring (6MR) windows ca. 3.4 Å in diameter, which run through the body diagonals of the unit cell, and six smaller four-membered ring (4MR) windows of ca. 0.8 Å which lie in the faces of the unit cell (Fig. 1)[15,16]. One reported gas-loading study in ZIF-8 used neutron powder diffraction (NPD) to analyse the adsorption process, and was used to determine the location of six adsorption sites for $D_2$[17,18]. Loading issues for $CD_4$ inhibited data

quality preventing the refinement of both the framework and guest molecules. This is a common problem in the study of MOFs and is attributed to a number of factors which include the large void space and weakly scattering guests. These issues make the task of experimentally locating and quantifying adsorption sites with a high degree of confidence challenging[19,20].

An alternative to the gas capillary cell is the diamond anvil cell (DAC), a small extreme-pressure device composed of two opposing diamond anvils which can apply GPa pressures to a pressure transmitting medium (PTM) which in turn transmits hydrostatic pressure to a crystalline sample. PTMs are generally liquids or soft solids which, when used to compress porous materials can lead to negative linear compressibility (e.g. in cyanide-bridged frameworks)[21,22], pressure-induced amorphisation (as observed in ZIFs)[23], and pressured-induced phase transitions, as observed in the high-pressure study of ZIF-8[13]. This last study was the first report to show that ZIF-8 undergoes a displacive phase transition as a result of the PTM penetrating into the pores on increasing pressure (herein, the ambient pressure and high-pressure structures are referred to as ZIF-8-AP and ZIF-8-HP, respectively). The displacive phase transition occurred at 1.47 GPa and is characterised by a re-orientation of the 4MR and 6MR imidazole rings, to form a gate open structure (Fig. 1).

The crystallographic determination of the transition to ZIF-8-HP became instrumental in interpreting the adsorption mechanism in ZIF-8, with the ZIF-8-HP phase being used as a model for in-situ powder diffraction data collected on ZIF-8 with included $N_2$ at more modest pressures (40 kPa)[24]. This experiment confirmed that the step in the $N_2$ adsorption isotherm at 77 K was induced by the same structural re-arrangement seen in the previous high-pressure study (to ZIF-8-HP). Grand canonical Monte Carlo (GCMC) simulations of an $N_2$ adsorption isotherm in ZIF-8-HP also showed a better agreement in the high-pressure region of the isotherm than with ZIF-8-AP. This demonstrated

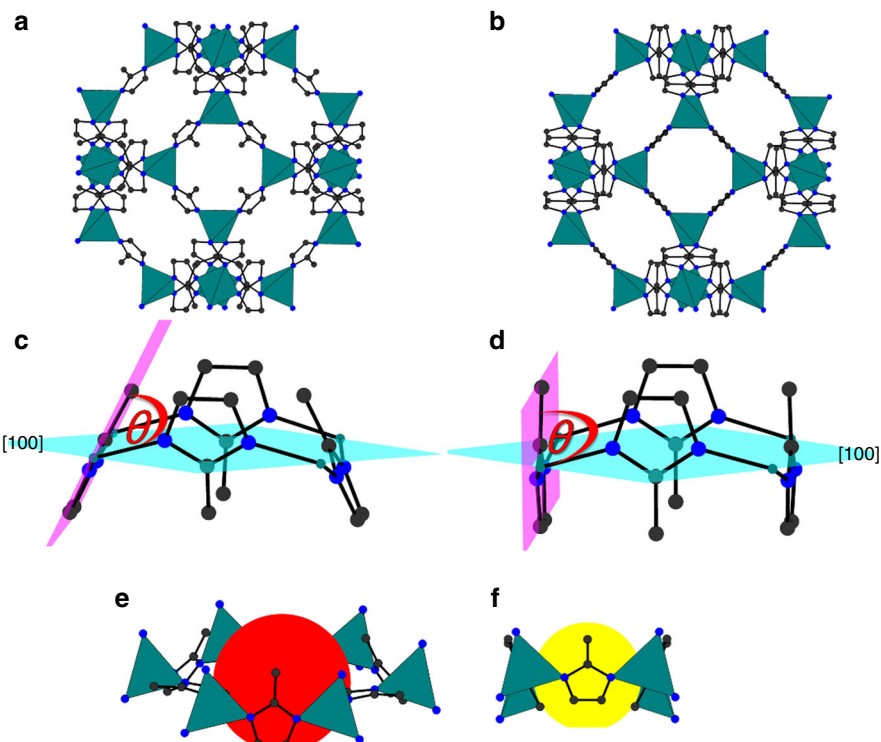

**Fig. 1** Structural differences of ZIF-8-AP and ZIF-8-HP. **a** ZIF-8-AP with 4MR window in the faces and 6MR windows in the body diagonal. **b** ZIF-8-HP with mlm linkers rotated by 30° compared to the AP structure **c** 4MR window showing the opening angle of 68° for ZIF-8-AP, **d** 4MR window showing the opening angle of 89° for ZIF-8-HP, **e** 6MR window with pore diameter of 3.0 Å and **f** 4MR window with pore diameter of 0.8 Å

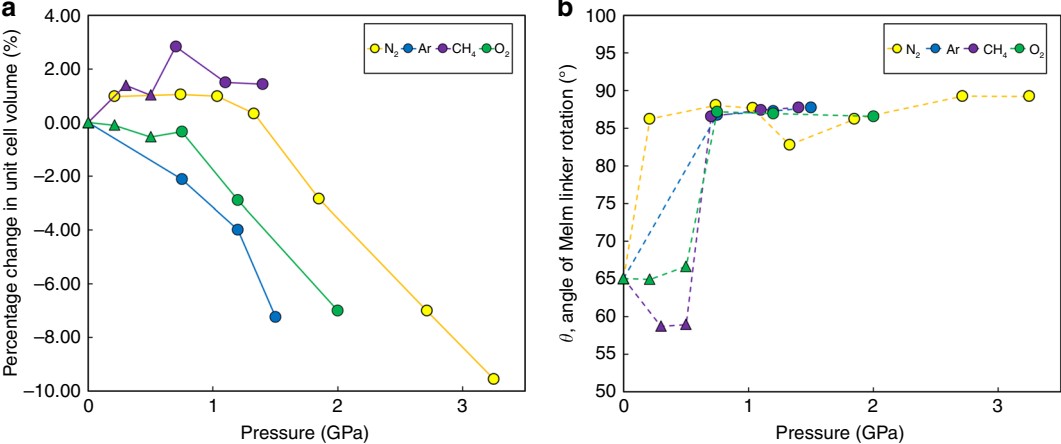

**Fig. 2** Compression and linker rotation in ZIF-8. **a** Change in unit cell volume of ZIF-8 with pressure for $N_2$ (yellow), $CH_4$ (red), Argon (blue), $O_2$ (green); **b** Change in angle of rotation of MeIm linker, $\theta$, in ZIF-8 upon increasing pressure for $N_2$ (yellow), $O_2$ (green), Ar (blue) and $CH_4$ (magenta). Triangles and circles indicate ZIF-8-AP and ZIF-8-HP phases, respectively

### Table 1 Effect of pressure on ZIF-8 for different gases

| Pressure CH₄ (GPa) | Unit cell volume (Å³) | % change in uc volume | SAV (Å³) | θ (°) | 4MR diameter (Å) | 6MR diameter (Å) |
|---|---|---|---|---|---|---|
| 0.00 | 4924.5 (2) | 0.00 | 2497 | 65.1 | 0.8 (1) | 3.0 (1) |
| 0.30 | 4993.1 (1) | 1.39 | 2580 | 58.7 | 0.7 (1) | 3.1 (1) |
| 0.50 | 4974.9 (4) | 1.02 | 2551 | 58.9 | 0.7 (1) | 3.1 (1) |
| 0.70[a] | 5063.8 (4) | 2.83 | 2657 | 86.6 | 2.5 (1) | 3.3 (1) |
| 1.10[a] | 4998.5 (4) | 1.50 | 2710 | 87.4 | 2.5 (1) | 3.2 (1) |
| 1.40[a] | 4995.1 (3) | 1.43 | 2586 | 87.8 | 2.5 (1) | 3.2 (1) |

| Pressure O₂ (GPa) | Unit cell volume (Å³) | % change in uc volume | SAV (Å³) | θ (°) | 4MR diameter (Å) | 6MR diameter (Å) |
|---|---|---|---|---|---|---|
| 0.00 | 4924.5 (2) | 0.00 | 2514 | 65.1 | 0.8 (1) | 3.0 (1) |
| 0.21 | 4919.8 (5) | −0.10 | 2487 | 64.9 | 0.9 (1) | 2.9 (1) |
| 0.50 | 4898.2 (3) | −0.53 | 2457 | 66.7 | 0.8 (1) | 2.9 (1) |
| 0.75[a] | 4908.0 (4) | −0.33 | 2522 | 87.2 | 2.2 (1) | 3.6 (1) |
| 1.20[a] | 4782.0 (9) | −2.89 | 2356 | 86.9 | 2.2 (1) | 3.4 (1) |
| 2.00[a] | 4579.9 (6) | −7.00 | 2255 | 86.6 | 2.4 (1) | 3.1 (1) |

| Pressure N₂ (GPa) | Unit cell volume (Å³) | % change in uc volume | SAV (Å³) | θ (°) | 4MR diameter (Å) | 6MR diameter (Å) |
|---|---|---|---|---|---|---|
| 0.00 | 4924.55 (23) | 0.00 | 2497 | 65.1 | 0.8 (1) | 3.0 (1) |
| 0.21[a] | 4972.3 (4) | 0.97 | 2548 | 86.2 | 2.5 (1) | 3.4 (1) |
| 0.74[a] | 4976 (1) | 1.04 | 2525 | 88.0 | 2.3 (1) | 3.5 (1) |
| 1.03[a] | 4972.8 (9) | 0.98 | 2512 | 87.7 | 2.3 (1) | 3.5 (1) |
| 1.33[a] | 4940.8 (3) | 0.33 | 2402 | 82.8 | 2.0 (1) | 3.5 (1) |
| 1.85[a] | 4784.5 (6) | −2.84 | 2364 | 86.2 | 2.0 (1) | 3.5 (1) |
| 2.72[a] | 4579.3 (5) | −7.01 | 2223 | 89.3 | 2.2 (1) | 3.2 (1) |
| 3.25[a] | 4454.0 (3) | −9.55 | 2054 | 89.2 | 2.2 (1) | 3.0 (1) |

| Pressure Ar (GPa) | Unit cell volume (Å³) | % change in uc volume | SAV (Å³) | θ (°) | 4MR diameter (Å) | 6MR diameter (Å) |
|---|---|---|---|---|---|---|
| 0.00 | 4924.5 (2) | 0.00 | 2497 | 65.1 | 0.8 (1) | 3.0 (1) |
| 0.75[a] | 4820.6 (3) | −2.11 | 2458 | 86.7 | 2.5 (1) | 3.3 (1) |
| 1.20[a] | 4727 (5) | −3.99 | 2405 | 87.3 | 2.4 (1) | 3.3 (1) |
| 1.50[a] | 4567.9 (4) | −7.24 | 2256 | 87.7 | 2.4 (1) | 3.2 (1) |

Diameters of 4MR and 6MR calculated using the void analysis routine in Mercury, (grid spacing of 0.2 Å)[14] SAV solvent accessible volume, calculated using PLATON[37]
[a] The ZIF-8-HP phase

the power of using high-pressure crystallography in tandem with computational methods to understand adsorption behaviour in MOFs at more modest pressures[24].

Using liquefied gases as a PTM in a DAC is common in the fields of high-pressure physics and mineralogy[25,26]. By using liquefied gases as PTMs, GPa pressures can be reached, facilitating higher occupation of guest molecules inside the pores. This results in atomistic resolution of the position of included gas molecules. In contrast, some gas-loading capillary experiments reported previously for ZIF-8[17,18] did not report any changes in the framework structure but did report difficulty modelling gas molecules, suggesting only modest levels of gas adsorption. By using GPa pressures, we can also ensure that we can override the particle size dependency associated with the ZIF-8 displacive phase transition[27,28]. Crucially, the DAC experiments can be carried out routinely using lab sources, making them more

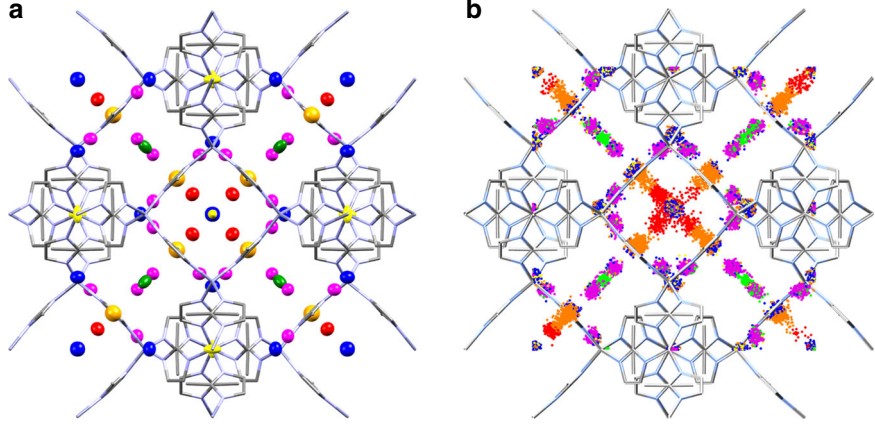

**Fig. 3** Comparison of Ar adsorption sites in ZIF-8-HP. **a** 1.20 GPa (298 K) crystal structure and **b** GCMC simulations at 100 kPa (at 83 K) of Ar. Colour scheme; Zn (light grey), N (light blue) and C (grey). Hydrogen atoms have been omitted for clarity. In **a** and **b** framework atoms are drawn as capped sticks, while Ar atoms are coloured according to the six symmetry-independent sites. In **a** Ar atoms are drawn with anisotropic displacement parameters (50% probability), while in **b** Ar atoms are shown as the binned positions with their relative energies from the GCMC simulation

accessible than synchrotron gas-cell experiments. Sotelo et al. recently demonstrated the power of liquefied gases as PTMs by loading $CH_4$ and $CO_2$ into the small pore MOF $Sc_2BDC_3$[29]. When the pressure of $Sc_2BDC_3$-$CH_4$ reached 2.50 GPa, hyper-filling of the framework with supercritical $CH_4$ was possible, allowing the first report of fully occupied $CH_4$ sites in the pores obtained at room temperature. In addition, loading of $CO_2$ to 0.2 GPa revealed a new adsorption site, which had never previously been observed, giving a much better agreement between the calculated uptake from diffraction measurements and experimental isotherm data[29].

Here, we present the first combined experimental and computational study of adsorption of small molecules (specifically $CH_4$, Ar, $O_2$ and $N_2$) into ZIF-8. By using the cryogenic loading method we were able to monitor structural changes in the framework upon uptake of these gases, experimentally locating the adsorption sites with high-precision in a MOF with much larger pores[29]. In addition, we sought to understand the framework responses and energetics of adsorption through plane wave density functional theory (DFT) and GCMC simulations. DFT calculations allowed us to probe the energy landscape of the framework geometry, whilst GCMC simulations of the adsorption process permitted quantification of the adsorption site energies.

## Results

**Effect of pressure on compressibility, geometry and loading.** The crystallographic data allowed changes in the framework compressibility, geometry and pore content to be monitored over the pressure range studied. As already highlighted in earlier reports[13], the orientation of the mIm linker [defined by the angle $\theta$ between the planes of the mIm atoms and the (100) crystallographic plane, see Fig. 1c, d] is particularly sensitive to pressure. Under ambient pressure and temperature conditions, $\theta$ is 65.1° and consequently the 4MR and 6MR window diameters measured 0.8 Å and 3.0 Å respectively. Upon sealing the DAC at the lowest possible pressures using $CH_4$ as a PTM (0.30 GPa), the unit cell volume of ZIF-8 increased by 1.39% (Fig. 2). This is indicative of the PTM penetrating into the pores of the framework (Table 1). The increase in electron density was accompanied by a decrease in $\theta$ to 58.7°, causing the diameter of the 4MR to decrease from 0.8 to 0.7 Å and the 6MR windows to increase from 3.0 to 3.1 Å. The calculated energy barrier to this small rotation was relatively modest, with an energy penalty of just 0.5 kJ mol$^{-1}$ per mIm linker (for more information on DFT calculations see

Supplementary Fig. 3). On increasing pressure to 0.50 GPa, the unit cell of ZIF-8-AP compressed (by 0.36%) while $\theta$ remained essentially unchanged (measuring 58.9°). Such a small change in rotation did not change the 4MR and 6MR window diameters, however the electron density within the pores steadily increased, equating to ca. 33 $CH_4$ molecules/uc (molecules per unit cell). At 0.70 GPa, the unit cell expanded further (by 2.83%), which was accompanied by a displacive phase transition to the ZIF-8-HP phase (Fig. 2), characterised by $\theta$ increasing to 86.6°, which caused the diameters of 4MR and 6MR to increase to 2.5 Å and 3.3 Å, respectively[13]. This was accompanied by a large change in solvent uptake into the pore (equating to ca. 89 $CH_4$ molecules/uc). The energy penalty for the framework to undergo such a large rotation was calculated to be 5.8 kJ mol$^{-1}$ per mIm linker. The energy of adsorption must, therefore, be greater to overcome this barrier. Increasing the pressure from 0.70 to 1.40 GPa led to compression of the HP phase, decreasing the unit cell volume by 1.40%, while $\theta$ continued to increase modestly, measuring 87.8° at 1.40 GPa (which corresponds to a 6.2 kJ mol$^{-1}$ penalty for rotation per linker). Conversely the 6MR diameter actually decreased in size between 0.70 and 1.40 GPa (by 2.40%) owing to the framework compression. Above 1.40 GPa, crystallinity deteriorated, and no structural data could be extracted.

Switching the PTM to $O_2$ revealed a similar trend in framework behaviour to that observed with $CH_4$ where there was a compression of the ZIF-8-AP phase marked by the decrease in four parameters: unit cell volume, $\theta$ and the diameters of 4MR and 6MR until the onset of the displacive phase transition to the ZIF-8-HP phase (marked by $^a$ in Table 1) when all four parameters increased. As the ZIF-8-HP phase was further compressed, the unit cell volume and the diameters of both 4MR and 6MR continued to decrease in size, while $\theta$ continued increasing (see Table 1). In the cases of $N_2$ and Ar as PTM, X-ray data is only available after the transition to ZIF-8-HP occurs. These PTM cause the 4MR, 6MR and $\theta$ to increase. More in depth information of how each particular gas affected the geometry of ZIF-8 with pressure can be found in Supplementary Note 1.

**Adsorption sites and energies in ZIF-8-HP.** In addition to monitoring the framework changes that occurred upon gas-loading, information pertaining to the location of adsorbed guest molecules in ZIF-8 could also be obtained, which stands testament to the quality of diffraction data that can be recorded when

using liquefied gases as PTMs in DACs. Gas loaded ZIF-8 crystal structures were modelled at 1.40, 0.75, 3.25 and 1.20 GPa in $CH_4$, $O_2$, $N_2$ and Ar PTM, respectively. These pressures were chosen as they met three key criteria: the electron density was at a maximum, the crystallographic data was the most complete and reflections were collected to the highest redundancy (Supple-

mentary Table 1). More details about how the X-ray data were refined can be found in Supplementary Methods. In addition, these crystallographic models and the ZIF-8-AP structure (with all guests removed) were used for GCMC simulations to model the gas uptake. The study of simulated isotherms is an area of active research, which allows theoreticians

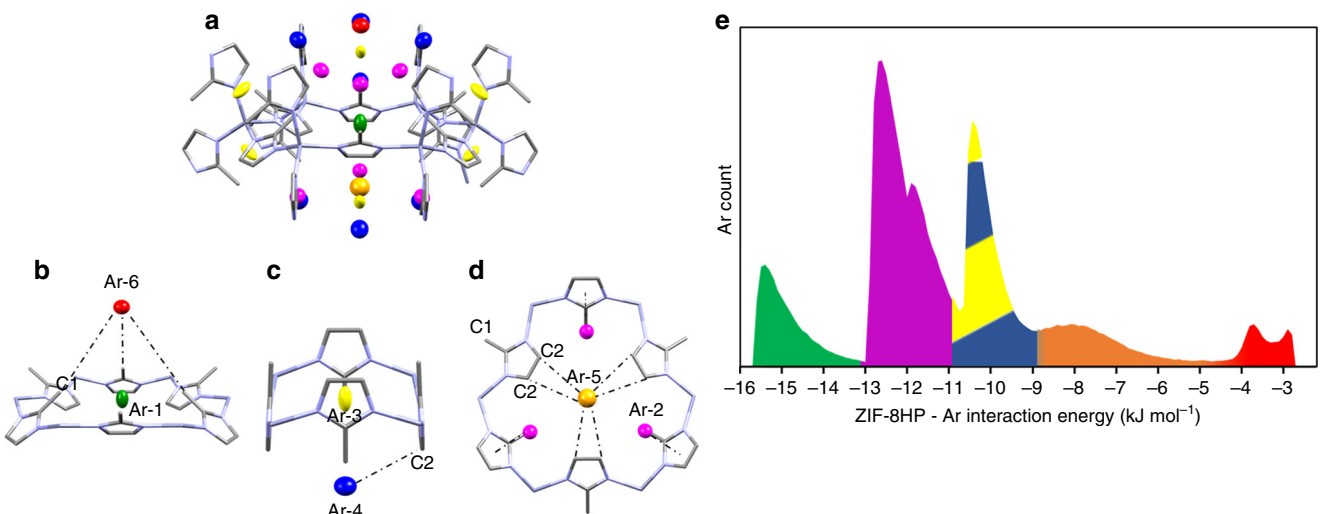

**Fig. 4** Crystallographic adsorption sites of Ar in ZIF-8 and their relative computed energies. **a** Ar adsorption sites in ZIF-8 at 1.20 GPa as shown on a 6MR window for Ar-1 (green), Ar-2 (red), Ar-3 (yellow), Ar-4 (dark blue), Ar-5 (orange), Ar-6 (red). Short contacts for **b** Ar-6 with the methyl C-atom (C1) and **c** Ar-4 with the imidazole C-atom (C2) and **d** Ar-5 (with C2) are drawn as black dotted lines and discussed in the text. **e** Histogram showing the frequency of guest-framework interaction energies at 100 kPa of guest during GCMC simulation in ZIF-8-Ar using ZIF-8-HP. Sections are coloured according to the adsorption sites Ar-1–Ar-6 (Ar-4 and Ar-5 cover the same energy range). Colour scheme; Zn (light grey), N (light blue) and C (grey). Hydrogen atoms have been omitted for clarity, ellipsoids for Ar-atoms are drawn at 50% probability, whilst the ZIF-8 framework is drawn as capped sticks

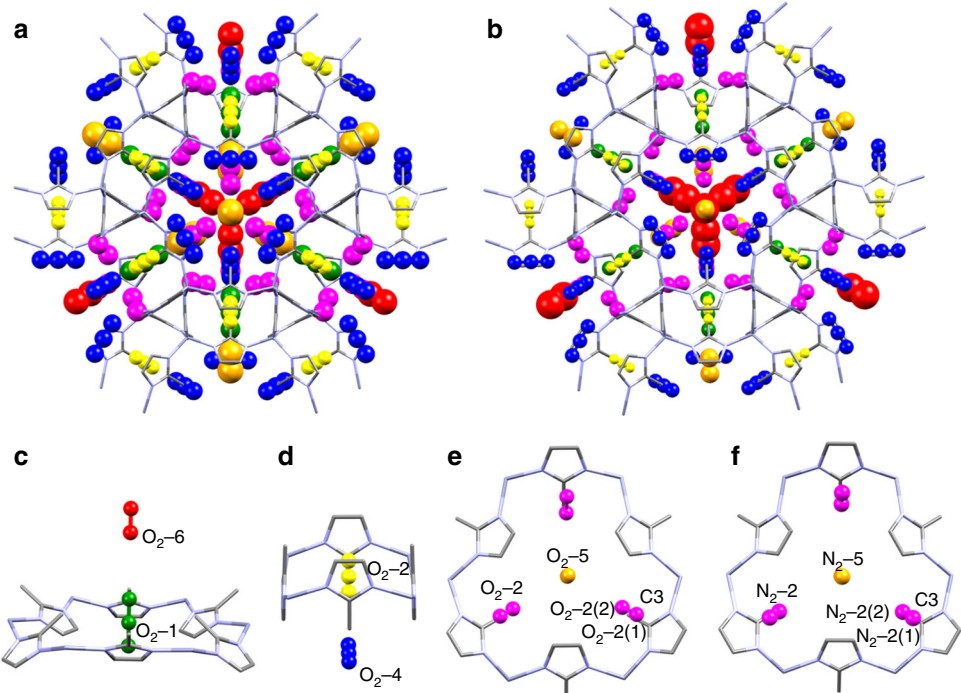

**Fig. 5** Experimentally-derived unit cell of ZIF-8-HP with $N_2$ and $O_2$ adsorption sites. **a** Adsorbed $N_2$ molecules at 3.25 GPa, and **b** adsorbed $O_2$ molecules at 1.20 GPa. In **a** and **b** guest shown as ellipsoids (50% probability) and each colour represents symmetry-independent adsorption sites. **c** 6MR window with sites $O_2$-1 and $O_2$-6. **d** 4MR window with sites $O_2$-3 and $O_2$-4. **e** 6MR window with $O_2$-2 and $O_2$-5 showing $O_2$(1)-$O_2$-(2)- C3 angle of 130°. **f** 6MR window with $N_2$-2 and $N_2$-5, showing $N_2$(1)-$N_2$-(2)- C3 angle of 150°. Note as **c** and **d** $N_2$ analogues are similar, their structures are not represented here

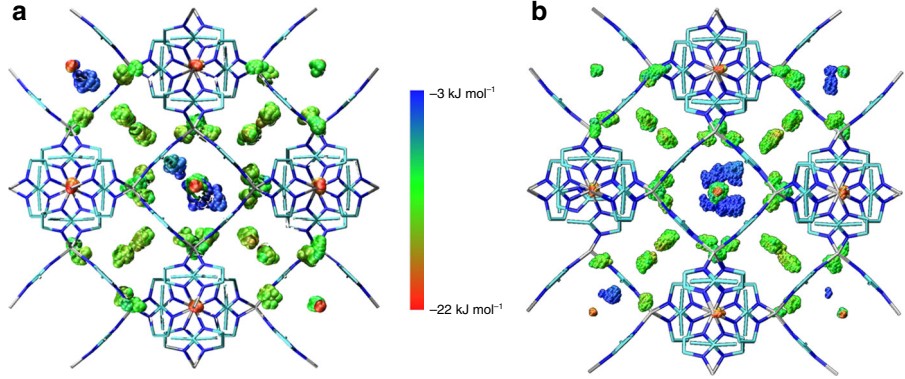

**Fig. 6** GCMC adsorption sites of $N_2$ and $O_2$. **a** Heat map of $N_2$ in ZIF-8-HP. **b** Heat map of $O_2$ in ZIF-8-HP. Points are centre of masses of diatomic molecules and the colour represents the framework–fluid interaction energy. Red corresponds to low energy (strong interaction) and blue to high energy (weak interaction)

to explain how structural changes affect the adsorption performance[16,30,31]. GCMC simulations allowed us to not only determine both the position of gas molecules, but, importantly, the energy of the sites.

When Ar was used as a PTM at 0.75 GPa, six symmetry-independent adsorption sites could be located within ZIF-8-HP, all of which were fully occupied. The GCMC simulation of Ar with the HP phase gave excellent agreement with those determined from the experimental crystal structure positions (Fig. 3).

Ar-1 (green, Fig. 4b) resides in the centre of each 6MR window and is the lowest energy (i.e. the most favourable) site, with a framework–argon interaction energy between ca. −16 to −13 kJ mol$^{-1}$ (Fig. 4e); Ar-2 (magenta, Fig. 4d) is the next most favourable site, and is located between the 4MR and 6MR windows, directly below every other mIm linker in the 6MR window (ca. −13 to −11 kJ mol$^{-1}$, Fig. 4e). The next two sites have comparable energies of ca. −11 to −9 kJ mol$^{-1}$: Ar-3 (yellow, Fig. 4c) is found in the centre of the 4MR window whilst Ar-4 (blue, Fig. 4c) sits below this point further into the nanopore. Ar-4 makes a close contact to the 4MR window being at a distance of 3.8118(1) Å from the imidazole C2 carbon (Fig. 4c). Adsorption site Ar-5 (orange, Fig. 4d) sits below the 6MR window equidistant from the mIm ligands, with the shortest contact distance (C2 to Ar-5) measuring 4.8677(1) Å; this long distance is reflected in a broad range of relatively weak energies from −9 to −4 kJ mol$^{-1}$ (Fig. 4e). Finally Ar-6 (red, Fig. 4b) sits in the centre of the large nanopore above the 6MR windows. The closest non-hydrogen atom contact to Ar-6 measures 5.9078(2) Å, from the methyl (C3) atoms (Fig. 4b). The simulations also confirm that this is a weak binding site, with interaction energies of the order of just ca. 3 kJ mol$^{-1}$.

Comparison of the energy histograms of framework–guest energies from the GCMC simulations for the ZIF-8-AP and ZIF-8-HP models (see Supplementary Fig. 6) show that the interaction energies are lowered (i.e. are more favourable) in the ZIF-8-HP phase. The difference between the guest-host interaction energies for the AP and HP phases varies between 3 and 7 kJ mol$^{-1}$ depending on the gas studied (further information on each gas interaction can be found in Supplementary Fig. 6). It is therefore clear that the transition to ZIF-8-HP is driven by the ability to form more favourable guest-host interactions, which, although is not surprising, is gratifying to confirm, but more importantly quantifies the energy gain involved on adsorption.

The position of the six adsorption sites found for Ar are also representative of those found in ZIF-8-HP for CH$_4$ (at 1.40 GPa); a detailed description of the energies of the adsorption sites for

this phase can be found in Supplementary Fig. 4, Supplementary Fig. 5 and Supplementary Note 2. Interestingly, this is the first single crystal X-ray structure of ZIF-8 loaded with CH$_4$ in the HP phase. Other studies, at lower pressures of 50 bar, have modelled the electron density of CH$_4$ within PLATON and their results agree with our ZIF-AP-CH$_4$ PLATON results (see Supplementary Table 1)[32].

In ZIF-8-HP at 3.25 GPa in $N_2$ and at 1.20 GPa in $O_2$ the adsorption sites are located in the same general positions as Ar, however, some of the sites exhibit positional disorder (Fig. 5). The most energetically favourable (i.e. the site which forms the strongest framework–guest interactions) $N_2$-1 and $O_2$-1 (green, Fig. 5) shows positional disorder perpendicular to the 6MR window, with one atom fully occupied, whilst the other was split across two positions (50% occupied). The shortest non-H contact (from the C2-atom on the mIm ligand) to the central N atom from $N_2$-1 measures 3.44(5) Å. The GCMC simulations agree with the diffraction data showing a range of positions above and below this plane (Fig. 6). $N_2$-2 and $O_2$-2 (magenta, Fig. 5) site shows no crystallographic disorder; it is positioned around a 3-fold axis that goes through the centre of the 6MR window, with each $N_2$-2 or $O_2$-2 sitting above three of the mIm linkers in the 6MR window. One clear difference between $O_2$-2 and $N_2$-2 is the angle which they make with the mIm linker, defined as the $X_2$-2 (1)-$X_2$-2(2)-C3 angle (where $X = N$ or O) (Fig. 5e,f). An angle of 90° would mean the molecule sits parallel to the mIm ring, however for $O_2$-2 the angle is 130° and the respective $N_2$-2 angle is 150°. Consequently it can be seen that $N_2$ will adopt an arrangement closer to an end-on intermolecular interaction (Fig. 5f) than a side-on interaction seen in $O_2$-2 (Fig. 5e).

$N_2$-3 and $O_2$-3 (yellow, Fig. 5) displayed similar disorder to $N_2$-1 and $O_2$-1, but the site was disordered around the plane of the 4MR window, with one fully occupied atom in the plane of the window with a 50% occupied atom above and below the plane. $N_2$-1 to $N_2$-3 and $O_2$-1 to $O_2$-3 are strongly interacting sites, where each diatomic molecule has comparable interaction energies ranging between −22 kJ mol$^{-1}$ and −12 kJ mol$^{-1}$ (Fig. 6). This broad range of energies comes from the orientation of the diatomic molecule with respect to the framework. For example in ZIF-8-$N_2$ the interaction energy is reduced if $N_2$-1 does not sit exactly in the plane of the 6MR. In addition, there is an energy penalty if the $N_2$-2 site is found closer to ZIF-8, decreasing from −14 kJ mol$^{-1}$ to −12 kJ mol$^{-1}$, as the distance shortens from 4.313 Å to 2.639 Å. From the diffraction data, $N_2$-4 and $O_2$-4 (blue, Fig. 5) were located below the 4MR window, exhibiting similar disorder to $N_2$-1/$O_2$-1 and $N_2$-3/$O_2$-3 and were found to have interaction energies in the region of −12 kJ

$mol^{-1}$ to $-9$ kJ $mol^{-1}$. On analysis of the multiple orientations calculated from the GCMC simulations, a wide spread of orientations below the 4-MR ring were observed, where $N_2$-4 and $O_2$-4 are located. This indicates that there is no preferred orientation for either of these pointing toward the methyl group or the C–C backbone of the mIm.

$N_2$-5 and $O_2$-5 (orange Fig. 5) situated below the 6MR window at a distance of 4.10(16) and 4.06(17) from C2, is in close contact with $N_2$-1 or $O_2$-1 with a contact distance of 1.9(3) Å and 1.8(4) Å, for $N_2$-5 and $O_2$-5, respectively. $N_2$-6 and $O_2$-6 (red Fig. 5) sits on the 4-fold axis of rotation going through the 6MR window. The closest guest-framework distance ($X...$C3, where $X =$ N or O) measures 5.3(2) Å, whilst the closest guest–guest is much shorter measuring 3.8(4) Å and 3.0(2) Å, for $N_2$-6...$N_2$-1 and $O_2$-6...$O_2$-1, respectively. The close contact of these sites with other adsorption sites and the significant distance from the framework may be an indication that these sites are only favourable due to guest–guest interactions rather than interactions with the framework. The GCMC simulations, which inherently have no symmetry imposed on the simulation box unlike the high symmetry ($I\bar{4}3m$) imposed in the crystallography, revealed three additional key pieces of information about these sites ($N_2$-5, $N_2$-6, $O_2$-5 and $O_2$-6). Firstly, the positions for these sites were scattered in the pore over the course of the simulation suggesting a flat energy landscape in the centre of the pore. Secondly, the sites were not present at all symmetry equivalent locations throughout the unit cell and lastly, the sites had very weak interaction energies with the framework, around $-3$ kJ $mol^{-1}$.

In conclusion, for the first time, high resolution high-pressure single crystal X-ray diffraction studies were combined with GCMC simulations and DFT calculations to understand gas adsorption behaviour. By using a cryogenic method of loading with a DAC, extreme pressures were used as a tool to force liquefied gases into the framework, thus building up occupancy of the gas in the framework. Using high-pressure crystallography, $O_2$, $N_2$, Ar and $CH_4$ were successfully loaded into ZIF-8 and used to determine the adsorption sites inside the framework at room temperature. The wide variety in behaviour confirms the suggestion, in previous crystallographic work by the group and others, that PTMs play a dynamic role in high-pressure studies of MOFs[13,33–36]. The energies of these crystallographically determined sites were calculated with GCMC simulations. The simulations helped explain a number of unanswered questions from the crystallographic data, including the hierarchy of adsorption sites, the low occupancies observed for some sites and the disorder of the guest molecules. The simulations also gave valuable information to confirm the orientation of the molecules in the pores. This work highlights the necessity of combining high-quality experimental X-ray data with computational methods. With this combined approach, we can monitor the structural changes in a MOF upon uptake of gases, and with theory calculate which interactions are the most favourable. These studies complement conventional adsorption studies by providing a detailed picture of the adsorption mechanism which is essential to understanding the adsorption process of flexible porous materials and their use in practical applications.

## Methods

**Synthesis of ZIF-8**. A solid mixture of zinc(II) nitrate hexahydrate (0.525 g, $1.76 \times 10^{-4}$ mol) and 2-methylimidazole (mIm; 0.015 g, $1.83 \times 10^{-4}$ mol) was dissolved in DMF (9 mL) in a 12 mL Teflon-capped vial which was heated at a rate of 200 °C $h^{-1}$ to 130 °C, held at this temperature for 24 h, and then cooled at a rate of 5 °C $h^{-1}$ to room temperature. Colourless polyhedral crystals were filtered from the reaction mixture, washed with methanol (3 × 5 mL), and dried in air (30 min).

**High-pressure cryogenic loading**. Each gas was cryogenically loaded into a DAC using the following basic procedure. A single crystal of ZIF-8 together with a chip of ruby (for pressure calibration) were loaded in a modified Merrill–Bassett DAC with 600 μm culet diamonds and a tungsten gasket[37,38]. Springs were placed on the pins of the DAC and a calibration performed to determine the open and closed positions of the DAC. The device was then placed inside a cryogenic gas-loading chamber (see Supplementary Fig.2), in the calibrated closed position and the chamber placed in a bath of liquid $N_2$ to equilibrate to 77 K. The gas ($N_2$, $O_2$, Ar or $CH_4$) was purged through the chamber until condensation occurred. The DAC was then opened to a pre-calibrated open position whilst in the bath of liquefied $N_2$ and the sample chamber exposed to the condensed gas for approximately 30 seconds before closing. The DAC was then removed from the bath and allowed to warm to room temperature before the pressure inside the cell was measured using the ruby fluorescence method[37].

**High-pressure X-ray diffraction**. High-pressure diffraction studies were collected on a Bruker APEX II diffractometer with graphite-monochromated Mo $K_\alpha$ radiation (0.71073 Å). Each gas loaded sample was studied over the following pressure regimes: $CH_4$ loaded in ZIF-8 (ZIF-8-$CH_4$) from 0.30 GPa to 1.40 GPa, $O_2$ loaded in ZIF-8 (ZIF-8-$O_2$) from 0.21 GPa to 2.00 GPa, $N_2$ loaded in ZIF-8 (ZIF-8-$N_2$) from 0.21 GPa to 3.25 GPa and Ar loaded in ZIF-8 (ZIF-8-Ar) from 0.75 GPa to 1.50 GPa. Upon increasing pressure further, sample deterioration resulted in loss of resolution. Structure refinements were carried out to the maximum resolution of each sample as determined from the intensity statistics. Data were collected for each pressure point in ω scans in eight settings of 2θ and ϕ, based on the strategy of Dawson et al. with an exposure time per frame of forty seconds and a step size of 0.5°[39]. The data were integrated using dynamic masks (generated using the programme ECLIPSE), in order to avoid regions of the detector shaded by the DAC, while the absorption corrections for the DAC and sample were carried out using the programme SADABS[40,41]. The data were then merged in XPREP[42].

**Crystal structure refinements**. Structure refinements were carried out in CRYSTALS starting from the ambient pressure structure of Park et al., (refcode VELVOY)[12,43]. All framework non-hydrogen atoms were refined anisotropically with thermal similarity and vibrational restraints applied to all non-hydrogen atoms except Zn. The SQUEEZE algorithm was applied (probe radius 1.2 Å, grid spacing 0.2 Å) to calculate the electron density in the pores and give an estimate to the number of guest species in the pore as a function of pressure[44]. The number of guest molecules was corrected for the residual electron density in the ambient pressure data set, where the crystal had been heated to 80 °C and exposed to vacuum for 12 h. The adsorbed gases were refined with crystallographic models at 0.75 GPa for Ar, 1.40 GPa for $CH_4$, 3.25 GPa for $N_2$ and 0.75 GPa for $O_2$. For the adsorbed gases, Ar and C atoms (for methane) were refined with anisotropic displacement parameters, whereas $O_2$ and $N_2$ were refined isotropically with distance, thermal and vibrational restraints applied. This is due to the increased number of parameters which were required to be refined against the data.

**Limiting window diameter analysis of the 4MR and 6MR windows in ZIF-8**. Each crystallographic structure was analysed in Mercury using the void analysis tools to determine the limiting pore diameter[14]. Guest molecules, if present in the pore, were removed before void analysis. The grid spacing was set to 0.2 Å and the probe size was increased until the 4MR windows were no longer accessible to solvent. This probe size diameter corresponds to the largest sphere that can be inserted without overlapping the framework atoms. The process was repeated for the 6MR windows of ZIF-8.

**Density functional theory single point energy calculations**. All calculations were performed using the CASTEP (version 5.11) simulation package[45]. The Hamiltonian operator was approximated using the Perdew-Burke-Ernzerhof (PBE) exchange-correlation functional augmented with the Tkatchenko-Scheffler dispersion correction[46,47]. The molecular wavefunction description was provided by 'on-the-fly' pseudopotentials and a plane wave basis set operating at 650 eV, which gave convergence to within 4 meV per atom. The electronic structure was sampled at the gamma position only in the Brillouin zone due to the large size of the primitive unit cell (resulting in a k-point sampling grid of no greater than 0.06 Å$^{-1}$). The ambient pressure crystal structure of ZIF-8 was fully optimised without any symmetry constraints to allow both the relaxation of the atomic positions and the unit cell parameters. The potential energy surface was searched for energy minima by means of the Broyden-Fletcher-Goldfarb-Shanno (BFGS) algorithm[48]. The structure was considered to be optimised when the energy per atom, maximum force, maximum stress and maximum atomic displacement converged to the values of 0.02 meV/atom, 0.05 eV/Å, 0.1 GPa and 0.002 Å, respectively. Once optimised, the coordinates of the methylimidazole linkers, defined as the angle θ between the planes of the mIm atoms and the (100) crystallographic plane [see Fig. 1c], were rotated through five degree increments up to 30° away from the ambient pressure structure coordinates; at each interval single point energy calculations were carried out at the same level of theory as the geometry optimisation. The data were then fitted to a second order polynomial to interpolate data between the 5° rotations.

**Grand canonical Monte Carlo simulations**. Gas adsorption was simulated using grand canonical Monte Carlo (GCMC) simulations[49], implemented in the multi-purpose code MuSiC[50]. The simulations were carried out using atomistic models of the frameworks ZIF-8-AP[12] and ZIF-8-HP[13]. The framework atoms were fixed at their crystallographic positions. At each pressure, $5 \times 10^7$ Monte Carlo steps were performed where each step consists of either a random translation, insertion or deletion, and, for $O_2$ and $N_2$, random rotation. The first 40 % of the steps were used for equilibration and the remaining used to calculate the ensemble averages. Standard Lennard-Jones (LJ) potentials were used to model the dispersive interactions between the framework and gases. The parameters for the force field were obtained from previous work from Fairen-Jimenez, who used a modified version of UFF (known as UFF*)[16]. Mulliken charges were used as the partial charges on the framework—the advantage of using these charges is that they were derived directly from the periodic DFT calculations, and, thus, capture the periodic nature of the MOF. The TraPPE force field was used to model $O_2$, $CH_4$ and $N_2$[51], while Ar was modelled using LJ parameters fitted to vapour—liquid data[52]. Coulombic interactions were included for $O_2$ and $N_2$ and calculated using Wolf Coulombic summations[53]. Interactions beyond 18 Å were neglected. To calculate the gas-phase fugacity the Peng–Robinson equation of state was used[54]. To gain better statistics for analysis of the positions of $O_2$ and $N_2$ in ZIF-8-HP, NVT MC simulations were performed with the same parameters and a loading corresponding to 100 kPa of $O_2$ or $N_2$ pressure.

**Data availability**. The X-ray crystallographic coordinates for structures reported in this study have been deposited at the Cambridge Crystallographic Data Centre (CCDC), under deposition numbers 1579394-1579417. These data can be obtained free of charge from The Cambridge Crystallographic Data Centre via "http://www.ccdc.cam.ac.uk/data_request/cif" www.ccdc.cam.ac.uk/data_request/cif". All other data underpinning this work is freely available via "http://hdl.handle.net/10283/3047"

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

## Acknowledgements

C.L.H. thanks the EPSRC and the University of Edinburgh for a studentship. C.A.M. acknowledges the UK Car-Parrinello consortium (EP/P022790/1) for access to the high performance computing resource ARCHER, as managed by the Edinburgh Parallel Computing Centre (EPCC). M.F. acknowledges EPSRC grant number EP/J003999/1, and studentship funding from EPSRC grant number EP/G03673X/1. S.A.M. thanks the EPSRC for EP/K033646/1 and EP/N01331X/1. T.D. acknowledges funding from the European Research Council (ERC) under the European Union's Horizon 2020 research and innovation programme (grant agreement No 648283).

## Author contributions

S.A.M., T.D. and C.A.M. conceived the project. C.L.H. synthesised the materials, conducted high-pressure crystallographic experiments. C.L.H. and C.W. conducted the cryogenic experiments. M.F. and K.K. designed the cryogenic experiment. C.L.H. and M.J.L. performed the Monte Carlo simulations. C.L.H. performed the DFT calculations. All authors discussed the results and commented on the manuscript.

## Additional information

**Competing interests:** The authors declare no competing interests.

