## [Peer Review File · Nature Communications]

Reviewers' comments:

Reviewer #1 (Remarks to the Author):

The manuscript by Hobday et al reports a comprehensive high pressure crystallographic investigation of a well-know MOF material, ZIF-8, with gas loading (N₂, O₂, Ar and CH₄). The structure determination at varying pressure and the locations of the gas molecules within the pores of ZIF-8 have been supported by GCMC/DFT modelling as appropriate. The loading of the gas into the crystal was achieved via a recently reported cryogenic technique (ref 25). The experiments were conducted with great care and presentation is clear. I enjoyed reading the paper. However, the manuscript in its present form is far too premature for consideration for publication. I highlight a few comments for authors to consider:

1. Abstract "The quality of crystallographic data collected

24 offers much greater atomistic detail compared to other gas-loading methods." I am not convinced by this at all, not least because there is no comparison between the present study (on locations of gas molecules in ZIF-8) and those reported previously using in-situ gas cell methods. Secondly, Table 1, the refined distances are subject to large errors (0.1 Å), leading to uncertainties of host-guest chemistry. I had a quick look at the R factors in the CIFs, and they are no smaller than those reported from gas-cell work. I am particularly interested in learning further on the CH₄ sites. Do the findings in this study improve any understanding based upon previous work as mentioned in the introduction? Therefore, to support this central claim, more/clear evidence is needed.

2. The discussion of changes (0.1Å) on the pore diameters (such as line 212) is not critically correct given the large errors from the refinement (0.1Å). This needs to be checked throughout. Whenever possible, the authors should include esd values in the main texts when describing their results, so one can better judge which difference is significant. I would not suggest to describe a pore with a size of 0.7-0.8Å. is this really a pore?

3. I am confused by the results reported in Table 1 and those in the CIFs. Firstly, the liquid density of the selected gas shows the trend of Ar>O₂>N₂>CH₄. In the field of gas adsorption in porous materials, at near saturation conditions (such as in this case under extremely high pressure), the max uptake of a certain gas should be proportional to the density of its liquid phase. Table 1 shows that the trend of gas uptake follows CH₄> O₂>N₂>Ar. This is unusual and needs justification. Secondly, the refined site occupancies in the CIFs do not really match those reported in Table 1. For example, the N₂-loaded ZIF-8 at 3.25Gpa, 32 N₂ per UC is expected; whereas in the CIF, 12x8.25= 99 is found. Some differences are allowed, but in this case, it is a 3 fold difference, which must be justified. This type of inconsistency is throughout the CIFs and those reported in Table 1. I suggest the authors to add a list of "found gas molecules per UC" in table 1 to cross check.

4. I am not certain what strategy was used to balance the refinement of site occupancy and thermal parameters. For example, in the CIF of N₂-loaded ZIF-8, the thermal parameters for N₂ molecules at different sites are notably different. Indeed, the difference can be as large as an order of magnitude. This is highly abnormal, in particularly for the similar studies using the gas-cell technique, and needs further consideration.

5. How accurate to extract the binding energy from GCMC calculations? I had believed only DFT calculation can afford reasonably accurate binding energies. As far as I understand, the GCMC-modelled sites in figure 3b show only the probability of site population, not the definite site positions. Related to this, figure 3 compares the experimental data at 1.20 GPa and modelling data at 1 bar (what temperature). They show a great degree of consistency, but what is the rational behind? looks to me, these two conditions are miles away between each other.

6. Minor error: there are some typos with refs (error! Reference source not found).

In conclusion, I think the manuscript has major gaps which need further attention before the publication can be considered.

Reviewer #2 (Remarks to the Author):

I have serious troubles with this manuscript that provides not only a biased interpretation of their results but neglects the contributions and important findings on this topic reported in the literature in the past 5 years.

First, some of the authors reported in 2011 the structural transition of ZIF-8 using computation methods and in-situ powder X-ray diffraction, after deformation at 1.47 GPa, postulating the swinging of the methyl imidazolium linker as the responsible for this behavior (1,2). Combined computation and experimental approaches have also been used to investigate the stepped-adsorption features of this material with different gas probes at cryogenic temperatures (3-5), and it is unacceptable to this reviewer that some of these works have been omitted in the citation list. The origin of the adsorption-induced deformation of ZIF-8 upon is an interesting topic, however there are some incoherencies in this manuscript. First of all, it has been demonstrated by in-situ XRD that the gas molecules loaded in the ZIF-8 structure do not induce a phase transition of the material itself, but rather a soft deformation. Indeed, ZIF-8 is a rigid material according to crystallographic data reported by several authors (3,7,8), thus the authors should be careful with the terminology used. ZIF-8 does not undergo a phase transition (crystallographic positions remained rather unchanged upon gas loading; the framework is just slightly deformed by the effect of the gas molecules confined in the host during the adsorption process, thus inducing the rotation of the linkers without modifying the lattice parameters. This is misunderstood in the manuscript.

Also, the (sic) "gas-loading experiments previously reported for ZIF-8" have reported that even modest levels of gas adsorption in the framework induce the deformation of the framework (5,6,7). Gas pressures lower than 1 kPa (at cryogenic temperatures) are enough to provoke the rearrangement of the linkers, similarly to compressing a fluid at 1.5 GPa (2). Thus, the authors should reconsider the terms "high pressure phase", provided that neither high pressure is needed, nor a phase transition of ZIF-8 occurs. This should be clarified, and contrasted with previous works available in the literature to avoid misinterpretations.

(1) Fairen-Jimenez, et al, Opening the Gate: Framework Flexibility in ZIF-8 Explored by Experiments and Simulations, *J. Am. Chem. Soc.*, 133 (2011) 8900.

(2) Moggach, et al, The Effect of Pressure on ZIF-8: Increasing Pore Size with Pressure and the Formation of a High-Pressure Phase at 1.47 GPa, *Angew. Chem.* 121 (2009) 121, 7221

(3) Ania et al, Understanding Gas-Induced Structural Deformation of ZIF-8, *J. Phys. Chem. C*, 3 (2012) 1159.

(4) Zhang, et al, Crystal-Size-Dependent Structural Transitions in Nanoporous Crystals: Adsorption-Induced Transitions in ZIF-8, *J. Phys. Chem. C*, 118 (2014) 118, 20727.

(5) Tanaka et al, Adsorption-induced structural transition of ZIF-8: A combined experimental and simulation study, *J. Phys. Chem. C*, 118 (2014) 8445. This is the only one cited in the manuscript

(6) Salas-Colera et al, Design and development of a controlled pressure/temperature set-up for in situ Studies of solid-gas processes and reactions in a synchrotron X-ray powder diffraction station, *J. Synch. Radiation* 22 (2015) 42.

(7) Tian et al, Role of Crystal Size on Swing-Effect and Adsorption Induced Structure Transition of ZIF-8, *Dalton Transactions* 45 (2016) 6893.

(8) Park, et al, Exceptional Chemical and Thermal Stability of Zeolitic Imidazolate Frameworks. *Proc. Natl. Acad. Sci. U.S.A.* 103 (2006), 10186–10191.

Reviewer #3 (Remarks to the Author):

The authors have pioneered a new approach to understanding the process of gas uptake in porous metal-organic frameworks (MOFs). By applying high pressures in the GPa range they have loaded the liquefied gases methane, argon, oxygen and nitrogen into the Zn-based framework ZIF-8. For the first time, an integrated study employing high pressure crystallography, grand canonical Monte Carlo simulations and periodic DFT calculations has detected the presence of (a) six symmetry-independent adsorption sites within the framework and (b) a transition to a high pressure phase. The crystallographic data was of sufficient quality to allow the determination of the structure to atomic resolution. GCMC simulations have allowed the different sites to be distinguished and classified in terms of their interaction energies. Finally, DFT calculations reveal the energy barrier for the transition between the ambient pressure phase and the high pressure one. This unique combination of the three techniques delivers a holistic approach to understanding the structural and energetic changes which attend the adsorption of small molecules into MOFs.

This approach has been successful and has produced significant insight into the adsorption process. The inclusion of small PTM molecules has allowed clear definition of all the adsorption sites, while the GCMC simulations have helped to resolve questions remaining after the crystal structure determination, notably the hierarchy of the adsorption sites and the low occupancies observed for some of them. The simulations also reveal the likely orientation of the PTM molecules within the pores.

I therefore recommend publication after addressing these minor considerations:

1. The authors correctly use GPa as the SI units of pressure but occasionally use bar. It is probably better to use 1 bar rather than 10^5 Pa.
2. In the caption to Figure 4, there is a definition of colours: "Colour scheme; Zn (grey), N (light blue) and C (grey)" which I do not understand. The text may be misplaced.
3. Tilde symbols are used to indicate approximate numerical values but I find that these are confusing where they precede minus signs: "ca." might be clearer.

Response to Reviewers' comments:

We appreciate your peer reviewed comments and have attached our thorough responses to your comments in a point by point manner in the document below:

Reviewer #1 (Remarks to the Author):

The manuscript by Hobday et al reports a comprehensive high pressure crystallographic investigation of a well-know MOF material, ZIF-8, with gas loading (N₂, O₂, Ar and CH₄). The structure determination at varying pressure and the locations of the gas molecules within the pores of ZIF-8 have been supported by GCMC/DFT modelling as appropriate. The loading of the gas into the crystal was achieved via a recently reported cryogenic technique (ref 25).

However, the manuscript in its present form is far too premature for consideration for publication. I highlight a few comments for authors to consider:

1. Abstract "The quality of crystallographic data collected 24 offers much greater atomistic detail compared to other gas-loading methods." I am not convinced by this at all, not least because there is no comparison between the present study (on locations of gas molecules in ZIF-8) and those reported previously using in-situ gas cell methods. Secondly, Table 1, the refined distances are subject to large errors (0.1 Å), leading to uncertainties of host-guest chemistry. I had a quick look at the R factors in the CIFs, and they are no smaller than those reported from gas-cell work. I am particularly interested in learning further on the CH₄ sites. Do the findings in this study improve any understanding based upon previous work as mentioned in the introduction? Therefore, to support this central claim, more/clear evidence is needed.

Response: *"I am not convinced by this at all, not least because there is no comparison between the present study (on locations of gas molecules in ZIF-8) and those reported previously using in-situ gas cell methods. Do the findings in this study improve any understanding based upon previous work as mentioned in the introduction? Therefore, to support this central claim, more/clear evidence is needed."* We thank the reviewer for

their comments, this led us to do a search of CSD (Cambridge Structural Database) of the gases studied in this paper, for gas loaded MOF structures. Of the gases studied in this paper, a CSD search highlighted that there were 7, 19, 27 and 24 crystal structures of MOFs containing Ar, CH₄, N₂ and O₂ – a total of 77 structures. Of these 77, 72 contained refined gas molecules within the pores. And of the remaining 72 structures, only 24 were carried out at room temperature. In relation to ZIF-8 alone, we therefore believe that the study we present is more comprehensive and detailed than anything else currently available in the literature, including the scope of gases studied and the methods used. Our studies are done at room temperature, therefore any changes in structure are caused solely by gas uptake as it is forced into the pores. We have added a figure to the SI, in SI-1, where we display a plot of the number of crystal structures with gases modelled in the pores from the past 20 years and discuss that this is still a field in its infancy and that the CSD and wider scientific community would benefit from more gas-loaded crystal structures in the database, as currently they only count for 0.08% of the MOF subset, and 0.007 % of the full CSD.

“Secondly, Table 1, the refined distances are subject to large errors (0.1 Å), leading to uncertainties of host-guest chemistry.” These are not refined crystallographic distance uncertainties, these are the errors associated with the method of determining the window diameters. Our crystallographic uncertainties are smaller than this, we describe many distances within the paper and also report their uncertainties. For example, see Page 13, Line 1: *“distance of 3.8118(1) Å from the imidazole C2 carbon”* and Line 3: *“(C2 to Ar-5) measuring 4.8677(1) Å”*. The least confident uncertainties we have are associated with guest molecules which are in the centre of the pore. Our simulations also show that these adsorption sites are more diffuse than those closer to the framework atoms, so larger uncertainties on the distances is not uncommon or unexpected

Reviewer one also commented that *“I had a quick look at the R factors in the CIFs, and they are no smaller than those reported from gas-cell work.”* The reviewer is correct, as we have said to another comment from Reviewer 1, we have done an extensive literature search of N₂, O₂, CH₄ and Ar gas included crystal structures. We have produced a plot of R-factor and year of publication, as well as R-factor and temperature, and placed our data on this plot. See supporting information section SI-1 and the Introduction in the main manuscript. Our data falls within the average.

However, for argon we do report the lowest R-factor for any MOF containing argon structure. R-factors within MOF data can vary depending on the size of the crystals and the size of the pores, where larger pores leads to less electron density to diffract and weaker high-angle reflections. We believe that all the data in this publication are of the highest quality we could obtain.

2. The discussion of changes (0.1Å) on the pore diameters (such as line 212) is not critically correct given the large errors from the refinement (0.1Å). This needs to be checked throughout. Whenever possible, the authors should include esd values in the main texts when describing their results, so one can better judge which difference is significant. I would not suggest to describe a pore with a size of 0.7-0.8Å. is this really a pore?

Response. These are not pore diameters, but window diameters. Whilst a change between 0.7 to 0.8 Angstroms does not seem large, our crystallographic data is within the certainty of being able to characterise the changes in window sizes (with errors on bond distances in the order of 0.01-0.03 Angstroms). The framework contains two windows into the central pore, namely a four-membered ring and a six membered ring. We have systematically monitored the window size as a function of pressure (and therefore gas uptake), by using a probe atom as as demonstrated in Table 1. The error of 0.1 Angstrom comes from the method of analysing the window size, where a 3D grid (of sample points separated by 0.2 Angstroms, which is the finest possible grid size possible with the available software - Mercury) is placed onto the framework and a probe atom is placed onto the grid to calculate the size of available volume. See manuscript section “Experimental and Computational Methods: Limiting window diameter analysis of the 4MR and 6MR windows in ZIF-8”. The probe atom is then increased in size until it can no longer fit into the windows. Placing a grid onto the structure to calculate pore sizes and limiting pore windows diameters is fairly common in the field of molecular simulation. Compare with – “*Linfeng et al. Nature Commun, 2017, 8, 1233 doi:10.1038/s41467-017-01166-3.*”

It is always good practise to have the error on these measurements, in some cases this is not always provided and we wanted to be as transparent as possible in our data accuracy.

3. I am confused by the results reported in Table 1 and those in the CIFs. Firstly, the liquid density of the selected gas shows the trend of $\text{Ar} > \text{O}_2 > \text{N}_2 > \text{CH}_4$. In the field of gas adsorption in porous materials, at near saturation conditions (such as in this case under extremely high pressure), the max uptake of a certain gas should be proportional to the density of its liquid phase. Table 1 shows that the trend of gas uptake follows $\text{CH}_4 > \text{O}_2 > \text{N}_2 > \text{Ar}$. This is unusual and needs justification. Secondly, the refined site occupancies in the CIFs do not really match those reported in Table 1. For example, the N_2 -loaded ZIF-8 at 3.25Gpa, 32 N_2 per UC is expected; whereas in the CIF, $12 \times 8.25 = 99$ is found. Some differences are allowed, but in this case, it is a 3 fold difference, which must be justified. This type of inconsistency is throughout the CIFs and those reported in Table 1. I suggest the authors to add a list of "found gas molecules per UC" in table 1 to cross check.

Response: We thank the reviewer for spotting our mistake in Table 1. We have carefully checked our manuscript, figures and tables and replaced any mistaken values. The column containing the pore content refers to the electron density calculated using the SQUEEZE algorithm in PLATON, corrected for the residual electron density in the evacuated ambient pressure structure. To clarify this, we have added a sentence to the experimental methods, crystal structure refinements section: "The number of guest molecules was corrected for the residual electron density in the ambient pressure data set, where the crystal had been heated to 80 ° C and exposed to vacuum for 12 hours." From the advice of the reviewer we have added our experimentally refined gas-included structures to Table One and have clarified where the occupancies come from by adding a sentence to the footer of table 1 "aAtoms or molecules per unit cell calculated from refined guest molecule content (otherwise estimated *via* PLATON SQUEEZE).

4. I am not certain what strategy was used to balance the refinement of site occupancy and thermal parameters. For example, in the CIF of N_2 -loaded ZIF-8, the thermal parameters for N_2 molecules at different sites are notably different. Indeed, the difference can be as large as an order of magnitude. This is highly abnormal, in particularly for the similar studies using the gas-cell technique, and needs further consideration.

Response: The isotropic displacement parameters, and refined occupancies are highly correlated parameters. It is unsurprising, that given the greater uptake of N₂ at specific sites, and lower occupancy at others (centre of the pore for example), that the thermal parameters are quite different. In the previous study by Zhang et al., the displacement parameters derived for included N₂ atoms for example are all collected at 115K or below. It is therefore unsurprising that our displacement parameters are quite different. Even by the authors own admission in this study, N₂ positions could not be determined at 293K for all but one site. This highlights the advantage here, in this study, where we have used pressure to force a range of gases inside the pores at room temperature, conditions far more likely to be used when separating or isolating gases from the atmosphere. Any changes caused by external pressure would have to be investigated by using a pressure transmitting media that does not penetrate the pores, or by potentially ball-milling, which has already been reported Cheetham et al., (*Chem. Commun.*, 2012,**48**, 7805-7807). Low-temperatures do have the added advantage that location of gas molecules is ‘easier’, due to reduced motion, and hence why the crystallographic data presented by Zhang et al., is so good. However, reducing temperature may also cause changes in the framework structure itself. Sc₂BDC₃ for example, undergoes a subtle phase transition from an orthorhombic *Fddd* phase, to a *C2/c* on cooling below 120K (Wright et al., *Inorg. Chem.*, **2011**, 50 (21), pp 10844–10858). We would argue that the data presented here, shows in a systematic way how the framework of ZIF-8 changes as a function of increasing gas content at room temperature, in unprecedented detail. In fact, in the entire MOF-subset of data included in the CSD, only 4 data sets include a model for N₂ at room temperature (see CSD refcodes LATFUI, QEVNAH, SEFTTOO AND SEFVOQ), the latter two coming from the study by Zhang et al., where only one N₂ site can be located. In this study, and in order to remove any bias or correlation across N₂ sites, thermal similarity and vibrational restraints were applied initially to all included N₂ atoms (i.e. all N₂ atoms were restrained to be the same). The occupancy of each N₂ site was then allowed to refine freely, before individual thermal parameters for N₂ sites were refined freely with no restraints, except for those to adjacent N-atoms. This has been clarified in the SI and text.

5. How accurate to extract the binding energy from GCMC calculations? I had believed

only DFT calculation can afford reasonably accurate binding energies. As far as I understand, the GCMC-modelled sites in figure 3b show only the probability of site population, not the definite site positions. Related to this, figure 3 compares the experimental data at 1.20 GPa and modelling data at 1 bar (what temperature). They show a great degree of consistency, but what is the rationale behind? looks to me, these two conditions are miles away between each other.

Response: The energies shown in this paper are *interaction* energies from GCMC calculations. Whilst DFT calculations are a higher level of theory, to perform GCMC simulations with DFT and not a classical force-field would be very time computationally expensive. Each pressure point in the GCMC simulation is composed of 5×10^7 Monte Carlo steps, where the energy is calculated after each step. This process is quick in classical GCMC, but would be very slow with DFT. In addition, our force field for the guest molecules are derived from vapour-liquid data, allowing for the best possible match with the adsorption measurement which occurs at vapour-liquid temperature.

Regarding Figure 3, where we compare our experimental data at 1.20 GPa (298 K) and modelling data at 1 bar (77 K), we are confident we can compare these two data points. Our experimental data is carried out on a large single crystal of ZIF-8 ~300 microns cubed. The barrier to diffusion on such a large crystal would be enormous. We use high-pressure as a tool to force the gas molecules into large single crystals, it is known that the phase transition is particle size dependent (Zhang et al. J. Phys. Chem. C 2014, 118, 20727–20733), so this technique is the only way to have single crystal data of such a transition which is its advantage. We have clarified the temperatures and pressures of these data points expressed in Figure 3 within the manuscript.

6. Minor error: there are some typos with refs (error! Reference source not found).

1. Response: We apologise, this was not included in the copy of the manuscript which was submitted and must have happened when we uploaded our manuscript. We have double checked the manuscript for all typos.

In conclusion, I think the manuscript has major gaps which need further attention before the publication can be considered.

Reviewer #2 (Remarks to the Author):

I have serious troubles with this manuscript that provides not only a biased interpretation of their results but neglects the contributions and important findings on this topic reported in the literature in the past 5 years.

First, some of the authors reported in 2011 the structural transition of ZIF-8 using computation methods and in-situ powder X-ray diffraction, after deformation at 1.47 GPa, postulating the swinging of the methyl imidazolium linker as the responsible for this behavior (1,2). Combined computation and experimental approaches have also been used to investigate the stepped-adsorption features of this material with different gas probes at cryogenic temperatures (3-5), and it is unacceptable to this reviewer that some of these works have been omitted in the citation list.

The origin of the adsorption-induced deformation of ZIF-8 upon is an interesting topic, however there are some incoherencies in this manuscript. First of all, it has been demonstrated by in-situ XRD that the gas molecules loaded in the XIF-8 structure do not induced a phase transition of the material itself, but rather a soft deformation. Indeed, ZIF-8 is a rigid material according to crystallographic data reported by several authors (3,7,8), thus the authors should be careful with the terminology used. ZIF-8 does not undergo a phase transition (crystallographic positions remained rather unchanged upon gas loading; the framework is just slightly deformed by the effect of the gas molecules confined in the host during the adsorption process, thus inducing the rotation of the linkers without modifying the lattice parameters. This is misunderstood in the manuscript. Also, the (sic) “gas-loading experiments previously reported for ZIF-8” have reported that even modest levels of gas adsorption in the framework induce the deformation of the framework (5,6,7). Gas pressures lower than 1 kPa (at cryogenic temperatures) are enough to provoke the rearrangement of the

linkers, similarly to compressing a fluid at 1.5 GPa (2). Thus, the authors should reconsider the terms “high pressure phase”, provided that neither high pressure is needed, nor a phase transition of ZIF-8 occurs. This should be clarified, and contrasted with previous works available in the literature to avoid misinterpretations.

- (1) Fairen-Jimenez, et al, Opening the Gate: Framework Flexibility in ZIF-8 Explored by Experiments and Simulations, *J. Am. Chem. Soc.*, 133 (2011) 8900.
- (2) Moggach, et al, The Effect of Pressure on ZIF-8: Increasing Pore Size with Pressure and the Formation of a High-Pressure Phase at 1.47 GPa, *Angew. Chem.* 121 (2009) 121, 7221
- (3) Ania et al, Understanding Gas-Induced Structural Deformation of ZIF-8, *J. Phys. Chem. C*, 3 (2012) 1159.
- (4) Zhang, et al, Crystal-Size-Dependent Structural Transitions in Nanoporous Crystals: Adsorption-Induced Transitions in ZIF-8. *J. Phys. Chem. C*, 118 (2014) 118, 20727.
- (5) Tanaka et al, Adsorption-induced structural transition of ZIF-8: A combined experimental and simulation study, *J. Phys. Chem. C*, 118 (2014) 8445. This is the only one cited in the manuscript
- (6) Salas-Colera et al, Design and development of a controlled pressure/temperature set-up for in situ Studies of solid–gas processes and reactions in a synchrotron X-ray powder diffraction station, *J. Synch. Radiation* 22 (2015) 42.
- (7) Tian et al, Role of Crystal Size on Swing-Effect and Adsorption Induced Structure Transition of ZIF-8, *Dalton Transactions* 45 (2016) 6893.
- (8) Park, et al, Exceptional Chemical and Thermal Stability of Zeolitic Imidazolate Frameworks. *Proc. Natl. Acad. Sci. U.S.A.* 103 (2006), 10186–10191.

Response to reviewer #2:

Firstly, we note that Reviewer #2 does not comment on any of the science carried out in this paper, only on our literature review within the introduction and nomenclature used.

The reviewer list eight citations within his review and mentions “*it is unacceptable to this reviewer that some of these works have been omitted in the citation list*”. In fact, from their review they believe we only cite one of these papers “(5) *Tanaka et al, Adsorption-induced structural transition of ZIF-8: A combined experimental and simulation study, J. Phys. Chem. C, 118 (2014) 8445. This is the only one cited in the manuscript*”, however we have cited

over half of the papers they believe we had not cited. Within the manuscript which reviewer #2 read, we have cited:

- Fairen-Jimenez, et al. J. Am. Chem. Soc., 133 (2011) 8900.
- Moggach, et al. Angew. Chem. 121 (2009) 121, 7221.
- Zhang, et al. J. Phys. Chem. C, 118 (2014) 118, 20727.
- Tanaka et al. J. Phys. Chem. C, 118 (2014) 8445.
- Park, et al. Proc. Natl. Acad. Sci. U.S.A. 103 (2006), 10186–10191.

We have taken on board their comments, however, and have included the remaining references they suggested, which include:

- Ania et al. J. Phys. Chem. C, 3 (2012) 1159.
- Salas-Colera et al. J. Synch. Radiation 22 (2015) 42.
- Tian et al. Dalton Transactions 45 (2016) 6893.

In response to reviewer #1, we have also extended the reference list to include a comprehensive review of gas filled MOFs within the CSD, not limiting ourselves to ZIF-8 alone.

Reviewer #2 also writes “*ZIF-8 is a rigid material according to crystallographic data reported by several authors (3,7,8), thus the authors should be careful with the terminology used. ZIF-8 does not undergo a phase transition (crystallographic positions remained rather unchanged upon gas loading; the framework is just slightly deformed by the effect of the gas molecules confined in the host during the adsorption process, thus inducing the rotation of the linkers without modifying the lattice parameters. This is misunderstood in the manuscript.*”

In our response to this, we refer to the International Union of Crystallography Dictionary definition of a phase transition: “*Phase transitions are classified on the basis of different criteria. Mechanistic classification (Buerger): reconstructive (with changes in the pattern of chemical bonds), displacive (characterized by only small atomic shifts), order-disorder (of the atomic distribution on given Wyckoff positions).* By definition, therefore, if the atomic positions change, a phase transition has occurred. In the case of ZIF-8, the transition is displacive, as the framework atoms move from the ambient positions. The unit cell

parameters do not necessarily have to change for a phase transition to occur. To avoid any confusion on the nature of the transition, we have now explicitly say that the transition is a displacive phase transition and refer to the IUCr dictionary definition.

ZIF-8 is a flexible material, (see Moggach, et al., *Angew. Chem.* 121 (2009) 121, 7221) and Fairen-Jimenez, et al., *J. Am. Chem. Soc.*, 133 (2011) 8900.), in fact it has been recently shown that isostructural ZIFs, ZIF-90 and ZIF-65, also show similar flexibility (Hobday et al., doi:10.1021/jacs.7b1089)

Reviewer #2 continues to write *“Thus, the authors should reconsider the terms “high pressure phase”, provided that nether high pressure is needed, nor a phase transition of ZIF-8 occurs.”* We call our ZIF-8-HP phase HP for high-pressure, in keeping with the original paper - see Moggach, et al, *The Effect of Pressure on ZIF-8: Increasing Pore Size with Pressure and the Formation of a High-Pressure Phase at 1.47 GPa*, *Angew. Chem.* 121 (2009) 121, 7221). This is a notation that has been widely adopted in the community see for example: Chokbunpiam et al. *J. Phys. Chem. C* 2016, 120, 23458–23468, Russell et al. *J. Phys. Chem. C*, 2014, 118 (49), 28603–28608, Tian et al. *Dalton Trans.*, 2016, 45, 6893-6900, Boutin et al. *J. Phys. Chem. Lett.* 2013, 4, 1861–1865. Coudert et al. *ChemPhysChem* 2017, 18, 2732 – 2738.

Reviewer #3 (Remarks to the Author):

The authors have pioneered a new approach to understanding the process of gas uptake in porous metal-organic frameworks (MOFs). By applying high pressures in the GPa range they have loaded the liquefied gases methane, argon, oxygen and nitrogen into the Zn-based framework ZIF-8. For the first time, an integrated study employing high pressure crystallography, grand canonical Monte Carlo simulations and periodic DFT calculations has detected the presence of (a) six symmetry-independent adsorption sites within the framework and (b) a transition to a high pressure phase. The crystallographic data data was of sufficient quality to allow the determination of the structure to atomic resolution. GCMC simulations have allowed the different sites to be distinguished and classified in terms of their interaction energies. Finally, DFT calculations reveal the energy barrier for the transition between the ambient pressure phase and the high pressure one. This unique combination of the three techniques delivers a holistic approach to understanding the structural and energetic changes which attend the adsorption of small molecules into MOFs.

This approach has been successful and has produced significant insight into the adsorption process. The inclusion of small PTM molecules has allowed clear definition of all the adsorption sites, while the GCMC simulations have helped to resolve questions remaining after the crystal structure determination, notably the hierarchy of the adsorption sites and the low occupancies observed for some of them. The simulations also reveal the likely orientation of the PTM molecules within the pores.

I therefore recommend publication after addressing these minor considerations:

1. The authors correctly use GPa as the SI units of pressure but occasionally use bar. It is probably better to use 1 bar rather than 10^5 Pa.
2. In the caption to Figure 4, there is a definition of colours: "Colour scheme; Zn (grey), N (light blue) and C (grey)" which I do not understand. The text may be misplaced.
3. Tilde symbols are used to indicate approximate numerical values but I find that these are confusing where they precede minus signs: "ca." might be clearer.

Response to reviewer #3: We thank this referee for their positive endorsement of our work. All minor considerations requested have been addressed.

Reviewers' comments:

Reviewer #1 (Remarks to the Author):

The authors have made some changes during the revision. I am satisfied with the answers to our earlier comments 2, 4 and 5. However, 1 and 3 will need further clarification.

1. If the authors are determined to claim this "The quality of crystallographic data collected offers much greater atomistic detail compared to other gas-loading methods.", solid evidence is required. Indeed, as shown in Figure S1, the R factors of the present study are in the average zone of the literature examples with Ar being smaller but N₂ higher. Thus, I am not convinced by this key claim, which clearly has an impact on a large number of studies of gas-loaded materials in literature.

Also I see no further discussion the CH₄ structures or the comparison with the state-of-the art in literature?

"Of these 77, 72 contained refined gas molecules within the pores. And of the remaining 72 structures, only 24 were carried out at room temperature. In relation to ZIF-8 alone, we therefore believe that the study we present is more comprehensive and detailed than anything else currently available in the literature, including the scope of gases studied and the methods used."

If there are 24 examples in the literature, what is the rationale for this study being the most comprehensive one? I do not quite get this. The authors should call out the discussion in the manuscript.

3. I am a little nervous with the results shown in Table 1. As it stands, the counts of molecules based upon X-ray structure and PLATON show rather large differences. For example, the CH₄ structures, at 1.10 GPa, PLATON suggested 113 CH₄ molecules in the UC, whereas X-ray structure showed 66 at a higher pressure of 1.40 GPa. Similar discrepancies are throughout the tables. This needs justification; my feeling is that something needs to be improved on the methodology applied.

Reviewer #2 (Remarks to the Author):

The points have been satisfactorily addressed

Response to Reviewers' comments:

Dear Reviewers,

We appreciate your peer reviewed comments and have attached our thorough responses to your comments in a point by point manner in the document below:

Reviewers' comments:

Reviewer #1 (Remarks to the Author):

The authors have made some changes during the revision. I am satisfied with the answers to our earlier comments 2, 4 and 5. However, 1 and 3 will need further clarification.

1. If the authors are determined to claim this "The quality of crystallographic data collected offers much greater atomistic detail compared to other gas-loading methods.", solid evidence is required. Indeed, as shown in Figure S1, the R factors of the present study are in the average zone of the literature examples with Ar being smaller but N₂ higher. Thus, I am not convinced by this key claim, which clearly has an impact on a large number of studies of gas-loaded materials in literature. Also I see no further discussion the CH₄ structures or the comparison with the state-of-the art in literature? "Of these 77, 72 contained refined gas molecules within the pores. And of the remaining 72 structures, only 24 were carried out at room temperature. In relation to ZIF-8 alone, we therefore believe that the study we present is more comprehensive and detailed than anything else currently available in the literature, including the scope of gases studied and the methods used." If there are 24 examples in the literature, what is the rationale for this study being the most comprehensive one? I do not quite get this. The authors should call out the discussion in the manuscript.

Response: In response to Reviewer 1's first comment, we have modified our abstract from "*The quality of crystallographic data collected offers much greater atomistic detail compared to other gas-loading methods*" to "*The cryogenic high-pressure loading method offers a new approach to obtaining atomistic detail on included guest molecules*".

We stand by our corrected statement and believe the method is versatile as it is easy to use with any gas that can be used as hydrostatic media – including gas mixtures. In addition, the method is straightforward to apply as all data collected for this manuscript was on a laboratory Bruker Apex II diffractometer equipped with only a Mo X-ray tube.

The reviewer also asks “*Also I see no further discussion the CH₄ structures or the comparison with the state-of-the art in literature?*”. We have included a comprehensive discussion of the CH₄ included ZIF-8 structure in the SI in section SI-5, along with detailed discussions of the simulations which accompany the experimental data. In addition, we have added a few sentences to the manuscript on the comparison with the state of the art CH₄ adsorption structures of ZIF-8 on page 14:

Interestingly, this is the first single crystal X-ray structure of ZIF-8 loaded with CH₄ in the HP phase. Other studies, at lower pressures of 50 bar, have modelled the electron density of CH₄ within PLATON and their results agree with our ZIF-AP-CH₄ PLATON results (see SI-5).” The reference for this study is: O. Shekhah, R. Swaidan, Y. Belmabkhout, M. du Plessis, T. Jacobs, L. J. Barbour, I. Pinnau and M. Eddaoudi, Chem Commun, 2014, 50, 2089-2092.

The referee seems to misunderstand our text on our study being comprehensive. In the manuscript we write “*In relation to ZIF-8 alone, we therefore believe that the study we present is more comprehensive and detailed than anything else currently available in the literature, including the scope of gases studied and the methods used.*” For ZIF-8 only, we claim to have the most comprehensive study in terms of guest-molecules studies and methods used. Please note our work is focussed not only on our novel gas-loaded crystallographic data, but also on the computational methods which we use. Both are very important in having a holistic understanding of adsorption in ZIF-8.

The reviewer then asks “*If there are 24 examples in the literature, what is the rationale for this study being the most comprehensive one? I do not quite get this. The authors should call out the discussion in the manuscript.*” as well expressing their point of view “*Thus, I am not convinced by this key claim, which clearly has an impact on a large number of studies of gas-loaded materials in literature*

Only 17 structures of gas molecules in all current MOF structures (of the $\approx 80,000$ in total in the CSD) have been carried out at room temperature. As a consequence, we think that our study on ZIF-8 is significant and impactful, as we increase the number of structures from 17 to 21, and our single study alone accounts for 20 % of all room temperature data. We have included a sentence in the SI which reads “Of the remaining structures, only 17 structures were collected at room temperature—3, 9, 3 and 2 for Ar, CH₄, N₂ and O₂, respectively. **Figure S1** shows for each gas the MOF structures as a function of publication year and R-factor and also temperature and R-factor.”

3. I am a little nervous with the results shown in Table 1. As it stands, the counts of molecules based upon X-ray structure and PLATON show rather large differences. For example, the CH₄ structures, at 1.10 GPa, PLATON suggested 113 CH₄ molecules in the UC, whereas X-ray structure showed 66 at a higher pressure of 1.40 GPa. Similar discrepancies are throughout the tables. This needs justification; my feeling is that something needs to be improved on the methodology applied.

Response: The PLATON results are derived in a different way to the modelled guest molecules, therefore, one would not expect their results to be identical. The methodology of how PLATON calculates residual electron density will include contributions from diffuse electron density scattering in which a single crystal atomistic model will not take into account. Moreover, if the overall electron density difference is $0.25 \text{ e}^-/\text{\AA}^3$ (a small difference), and the accessible void volume is $\sim 2500 \text{\AA}^3$, then of course this will amount to a large amount of electron density which will be taken into account in PLATON.

The PLATON analysis allows an understanding, as a first approximation, of how the electron density in the pores changes with increasing pressure, including the very diffuse electron density in the pores. We have included it here for completeness, and in order to help others reproduce the results if they wish. We have tried to make this clear in the text, by explaining how the routine has been used to give an estimate of the pore content, as stated in the manuscript on Page 6: “*The SQUEEZE algorithm was applied (probe radius 1.2 Å, grid spacing 0.2 Å) to calculate the electron density in the pores and give an estimate to the number of guest species in the pore*

as a function of pressure.”, and Page 11 (footer of table 1) *“Atoms or molecules per unit cell calculated from refined guest molecule content (otherwise estimated via PLATON SQUEEZE).”*

In an effort to simplify the reading of the text for the reader, we have removed the column displaying the molecules/atoms per unit cell in Table 1 in the manuscript, and have included the PLATON SQUEEZE results to SI, Table S1. In SI section SI-4 (Crystallographic details), which contains the PLATON SQUEEZE results, there is also a detailed explanation of the discrepancies between both methods. The section in the SI reads:

“For CH₄ and O₂ loaded ZIF-8, diffraction data collected before the phase transition could only be refined against the framework, as the gas molecules could not be modelled atomistically in ZIF-8AP, due to low guest content, as the electron density in the pores was too diffuse. As a consequence, the pore content was modelled using the SQUEEZE algorithm in PLATON for all pressure points in all PTMs for completeness. After the phase transition to ZIF-8HP, when the electron density in the pores was sufficient enough to model and the data quality was at its highest, structural models were obtained for both the framework and the included gas molecules at 1.40, 0.75, 3.25 and 1.20 GPa for CH₄, O₂, N₂, and Ar.

*The SQUEEZE algorithm was applied (probe radius 1.2 Å, grid spacing 0.2 Å) to calculate the electron density in the pores and give an estimate to the number of guest species in the pore as a function of pressure. The number of guest molecules was corrected for the residual electron density in the ambient pressure data set and is shown in **Table S1**, along with the crystallographically modelled number of guest molecules per unit cell. As PLATON calculates the residual electron density in a different way to the atomistic modelling of guest molecules with spherical form factors, one would not expect their results to be identical. The methodology of how PLATON calculates residual electron density will include contributions from diffuse electron density scattering which an atomistic model will not take into account. The algorithm is applied here to give an estimate of the number of guest molecules in the pore. It is also recognised as one of only a few available methods, and one which is regularly used as a ‘standard method’ for modelling guest molecules when electron density is too diffuse.³⁻⁵ For example, for each gas studied in Table S1, one can see*

that the general trend in molecules per unit cell is increasing with increasing pressure – a sensible result – however, the absolute value must be taken with some caution and we believe the atomistic refinements gives more accurate occupancies of guest molecules. The SQUEEZE data is contained in the SI for completeness and transparency of data.”

We hope that this discussion within the SI on our PLATON results versus the “true” crystallographic model will show that we are transparent in our data processing and hope that the readers find it useful information if they were to repeat the study.

Reviewer #2 (Remarks to the Author):

The points have been satisfactorily addressed.

Response: We thank reviewer #2.

REVIEWERS' COMMENTS:

Reviewer #1 (Remarks to the Author):

The authors have made sufficient revisions to the manuscript and addressed our earlier concerns, and I can now recommend for publication.